# Robust Intelligent Monitoring and Measurement System toward Downhole Dynamic Liquid Level

**DOI:** 10.3390/s24113607

**Published:** 2024-06-03

**Authors:** Zhiyang Liu, Qi Fan, Jianjian Liu, Luoyu Zhou, Zhengbing Zhang

**Affiliations:** 1School of Electronics Information and Electrical Engineering, Yangtze University, Jingzhou 434023, China; zhiyangliu@dingtalk.com (Z.L.); luoyuzh@yangtzeu.edu.cn (L.Z.); 2Institute for Artificial Intelligence, Yangtze University, Jingzhou 434023, China; 3Logging Technology Research Institute, China National Logging Corporation, Xi’an 710077, China; fanqi_cpl@cnpc.com.cn (Q.F.); zycjliujianj@cnpc.com.cn (J.L.)

**Keywords:** dynamic liquid level monitoring and measurement, noise reduction algorithm, Internet of Things, LoRa

## Abstract

Dynamic liquid level monitoring and measurement in oil wells is essential in ensuring the safe and efficient operation of oil extraction machinery and formulating rational extraction policies that enhance the productivity of oilfields. This paper presents an intelligent infrasound-based measurement method for oil wells’ dynamic liquid levels; it is designed to address the challenges of conventional measurement methods, including high costs, low precision, low robustness and inadequate real-time performance. Firstly, a novel noise reduction algorithm is introduced to effectively mitigate both periodic and stochastic noise, thereby significantly improving the accuracy of dynamic liquid level detection. Additionally, leveraging the PyQT framework, a software platform for real-time dynamic liquid level monitoring is engineered, capable of generating liquid level profiles, computing the sound velocity and liquid depth and visualizing the monitoring data. To bolster the data storage and analytical capabilities, the system incorporates an around-the-clock unattended monitoring approach, utilizing Internet of Things (IoT) technology to facilitate the transmission of the collected dynamic liquid level data and computed results to the oilfield’s central data repository via LoRa and 4G communication modules. Field trials on dynamic liquid level monitoring and measurement in oil wells demonstrate a measurement range of 600 m to 3000 m, with consistent and reliable results, fulfilling the requirements for oil well dynamic liquid level monitoring and measurement. This innovative system offers a new perspective and methodology for the computation and surveillance of dynamic liquid level depths.

## 1. Introduction

With the advancement of digital oilfield construction, the automatic acquisition and analysis of oil well dynamic liquid level parameters are crucial tasks that directly impact the enhancement of oil well production, reductions in extraction energy consumption and improvements in economic benefits [1,2]. The real-time adjustment of the pumping unit’s stroke rate based on the depth of the dynamic liquid level can significantly increase the production efficiency of the oilfield. When the dynamic liquid level depth decreases, it indicates a sufficient reservoir fluid supply, and increasing the stroke rate of the pumping unit can enhance the production per unit of time. Conversely, when the dynamic liquid level depth increases, it suggests the weakening of the reservoir’s fluid supply capacity. In such cases, reducing the stroke rate of the pumping unit or temporarily halting the machine can maintain production while simultaneously lowering the energy consumption per unit of time [3,4]. Therefore, the real-time monitoring and accurate measurement of the oil well dynamic liquid level depth hold significant importance.

Over the past two decades, experts have utilized various technologies and methods to design and develop a range of instruments for the monitoring and measurement of the dynamic liquid level depth in oil wells. Common methods include the float method, pressure gauge detection method, fiber optic sensing method, indicator diagram method and acoustic wave method [5,6,7,8,9]. In 2008, McCoy et al. developed a dynamic liquid level detection system, based on the acoustic wave method, that collected high-frequency coupling echoes and low-frequency liquid level echoes through dual channels. However, the system was unable to perform automatic signal analysis and calculations of the liquid level depth [10]. In 2013, Liu et al. used the explosive pressure from bullet ammunition as a means of acoustic wave emission. At the moment of the gunpowder explosion, the emitter housing releases a high-energy, low-frequency acoustic wave signal for dynamic liquid level depth measurement over a short period. This method generates acoustic waves with high energy and distinct echo signals, but the introduction of explosive materials significantly increases the risk factor [11]. In 2015, Li et al. utilized high-pressure gas injection through the casing to produce acoustic waves, requiring a certain pressure differential between the well casing pressure and atmospheric pressure. This method offers low measurement errors and simple operation, but it is only suitable for high-pressure wells with casing pressures ranging from 5 Mpa to 25 Mpa [12]. In 2017, Zhang et al. proposed a self-correlation analysis-based acoustic wave processing method for oil well dynamic liquid levels. This method is only applicable to shallow wells with multiple liquid level echoes, and it is not suitable for deep wells with weak acoustic signals [13].

Compared to previous downhole dynamic liquid level monitoring systems, the noise suppression algorithm that we have developed demonstrates a distinct advantage. Traditional monitoring methods often lack specialized algorithms to deal with periodic and random noise. During multiple downhole dynamic liquid level monitoring experiments, we observed that underground environmental factors, on-site construction processes and operators’ handling of the equipment could all introduce noise that interferes with the monitoring results of the liquid level depth. In particular, periodic noise in the downhole environment, such as the impact sounds produced by the up-and-down movement of the nodding donkey of the oil extraction machine, as well as random noise like sudden vehicle noise on-site, could be mistakenly identified as liquid level echo signals, thereby negatively affecting the monitoring outcomes. To effectively suppress this noise and enhance the liquid level echo signal, we have taken targeted measures. We have designed algorithms specifically to deal with periodic and random noise; they can accurately identify and reduce the noise interference while increasing the amplitude of the liquid level echo signal. With these advanced noise suppression techniques, we have significantly reduced the error in liquid level depth identification and improved the accuracy and reliability of the monitoring results. Our algorithm has optimized the performance of the monitoring system, enabling it to provide high-quality monitoring data even in complex downhole environments.

In practical applications at the Jianghan Oilfield, we conducted measurements of the dynamic liquid level depth in multiple oil wells and analyzed the measurement errors. Through targeted noise processing, we successfully controlled the measurement error within 2%, meeting the accuracy requirements for dynamic liquid level measurement. We will continue to refine our noise reduction algorithms to further improve the reliability and accuracy of our measurements.

This paper introduces an innovative intelligent monitoring system for the dynamic liquid levels in oil wells, designed to address the harsh conditions and complex noise environments typical of oilfields. The system features an automated denoising algorithm that accurately identifies the index value of liquid waves and rapidly calculates the subsurface acoustic velocity and liquid depth, significantly enhancing the accuracy and efficiency of the measurements. In terms of system development, it utilizes an electrically controlled spring piston to generate infrasound excitation, simplifying the structure and enhancing the safety. The system incorporates cross-platform visualization software based on the PyQt framework, enabling data analysis and visualization on Windows, Linux, and MacOS. Furthermore, the system employs LoRa and 4G communication modules for the real-time remote transmission of data and uses wireless modules to upload the data to cloud servers, thereby enhancing the storage capacity and enabling the comprehensive analysis of data from multiple wells. This improves the system’s interpretive power and monitoring efficiency, ultimately achieving the goal of automated, unattended monitoring.

The remainder of this work is organized as follows. Section 2 elaborates on the principles and methods of dynamic liquid level depth monitoring; Section 3 introduces the system design, including the architecture and noise reduction algorithms; Section 4 presents the on-site experiments and error analysis; Section 5 summarizes the work, discussing the system’s principles, practical applications, limitations and future directions.

## 2. Principles and Methods

### 2.1. Principle of Oil Well Dynamic Liquid Level Depth Measurement by Acoustic Wave Method

The principle of measuring the dynamic liquid level depth in oil wells using the acoustic wave method is based on the concept of acoustic ranging. After the acoustic wave generator at the wellhead produces an acoustic wave signal, the wave propagates downward along the annular space. Upon encountering a reflection interface formed by the coupling, a small portion of the wave generates a reflection (coupling wave), which is received by the wellhead receiver. The majority of the acoustic wave continues to propagate downward, with its energy gradually dissipating and the signal strength progressively diminishing. Eventually, a fraction of the wave reaches the liquid level, and the resulting reflection (liquid level wave) is received by the receiver. This liquid level wave exhibits significant differences from the wellhead transmission wave and the coupling reflection in both the time and frequency domains [14,15,16,17]. The propagation model of the acoustic wave in the tubing is illustrated in Figure 1, and the actual acoustic waveform and the positions of various reflections captured are shown in Figure 2.

Based on the aforementioned process of acoustic wave propagation, the formula for the calculation of the depth of the liquid level in the well is derived.
(1)$$H=\frac{vt}{2}$$
where H represents the depth of the liquid level, v denotes the propagation velocity of the acoustic wave within the well, and t is the time taken for the acoustic wave to reach the liquid level and for the reflected wave from the liquid surface to be received by the receiver. It is evident that the calculation of the dynamic liquid level depth primarily depends on the accurate determination of the propagation time of the liquid surface wave and the propagation speed of the acoustic wave in the well. The acquisition of time is mainly achieved through a liquid surface echo recognition algorithm, which will be introduced in Section 3. The propagation speed of the acoustic wave in the well will be calculated using the coupling wave from the echo signal.

### 2.2. Calculation of Sound Velocity Based on Coupling Waves

The coupling wave is formed by a small portion of the reflected wave generated when the acoustic wave encounters the reflective interface created by the oil tube coupling. Assuming that the length of the oil tube is $L_{0}$, and the travel path difference for the coupling wave echo is 2$L_{0}$, with the frequency of the coupling wave obtained through the Fourier transform being $f_{0}$, the formula for the calculation of the propagation speed of sound waves in the oil well is as follows [18,19,20,21,22,23,24,25].
(2)$$v=2L_{0}f_{0}$$

The position of the coupling on the oil tube is fixed; hence, it is straightforward to determine the length of the oil tube at the location of the coupling. There is a standardized specification for the oil tubes used in the wellfield, with the couplings measuring 9.8 m in length. To more conveniently observe and analyze the coupling wave reflections, the data points of the coupling wave reflections are extracted from Figure 3.

To mitigate the impact of noise, an appropriate Butterworth bandpass digital filter can be designed for the signal filtering of the coupling wave. Since the frequency of the coupling wave is around 15 Hz, a bandpass filter with cutoff frequencies at 10 Hz and 20 Hz is selected. The filtered result is shown in Figure 4.

After the filtering process, it is possible to effectively evaluate the number and frequency of the couplings, thereby reducing the interference of noise. Then, the filtered signal is subjected to a Fourier transform to obtain the spectrum. As shown in Figure 5, the dominant frequency of the coupling wave reflection signal can be estimated to be 17.73 Hz. Consequently, the propagation speed of the acoustic wave in the well can be calculated using Equation (2).

## 3. Design and Implementation

The development of monitoring and measurement systems in the industrial sector, based on Internet of Things (IoT) hardware and software technology, is an important trend. A large number of heterogeneous devices are interconnected, serving various engineering technology fields, such as smart energy, smart transportation, smart cities, smart agriculture and smart healthcare. To meet the requirements of end-to-end low latency, high availability and scalability and the capability to process massive amounts of data, it is necessary to design intelligent terminal software systems at the edge. At the edge, short-range communications (ranging from 1 m to 50 m) are achieved through LoRa or Bluetooth, followed by the aggregation function to the oilfield data center, completed through 4G/5G transmission systems [26,27,28,29,30,31]. As shown in Figure 6, this monitoring and measurement system is composed of acquisition devices, network transmission, cloud services, and software systems. The core module of the intelligent oil well dynamic liquid level monitoring and measurement system mainly consists of three parts, including the wellhead excitation and acquisition device, the liquid level echo depth measurement algorithm and the data visualization software.

### 3.1. Wellhead Excitation and Acquisition Device

The circuit of the wellhead excitation and data acquisition device comprises four parts, including the power supply circuit, the motor control circuit, the signal acquisition circuit and the data transmission circuit, as shown in Figure 7.

The microcontroller unit (MCU) controller is renowned for its high integration, low power consumption, programmability, user-friendliness and widespread application. It is primarily categorized into several main types, including MCUs based on 8-bit, 16-bit or 32-bit architectures, digital signal processors (DSPs) specifically designed for control systems and system-on-chip (SoC) solutions tailored to embedded applications.

The power supply circuit module is responsible for converting a 24 V DC input into a 5 V DC output, which facilitates the powering of various modules and chips, ensuring their normal operation.

The motor control module is operated by the MCU module, which issues control signals to achieve the forward and reverse rotation of the motor by switching the contacts of the relay, thereby completing the compression and release of the spring.

The signal acquisition module is tasked with gathering data from the sensors, which includes monitoring the voltage across different components. This module employs a dedicated A/D conversion chip, which leverages high-speed communication protocols for data transmission. It is capable of simultaneously acquiring data from three channels.

The data transmission module serves as the interface for data communication with the terminal, which could be a software program on an intelligent device or a remote server. The main controller utilizes serial communication protocols to transmit data to the terminal. Additionally, the terminal can issue control commands to the main controller for system configuration and adjustments.

### 3.2. Liquid Level Depth Measurement Algorithm

Based on engineering experience from oil extraction wells, the depth of the dynamic liquid level ranges from 500 to 3000 m, the sound speed in oil extraction wells is between 320 m/s and 360 m/s and the signal sampling rate is between 250 Hz and 1 kHz [32,33]. The single infrasound wave acquisition time for the dynamic liquid level monitoring system is 90 s, during which data are collected once every 4 milliseconds, resulting in a total of 22,500 data points. The collected signals from point 0 to 22,500 are sequences used to identify the subsurface noise signals. Therefore, using parameters such as the depth of the dynamic liquid level, sound speed and sampling rate, the 5000 points starting from the moment of infrasound wave excitation can be considered as the time sequence for the calculation of the depth of the liquid level wave.

Although it is relatively easy to obtain signals using the acoustic wave method, there are more challenges in processing these acoustic signals. Hence, the selection and comprehensive application of effective digital signal processing methods are particularly crucial. Due to the complex environment in oil wells, the actual acoustic signals are mixed with various types of noise, and the useful signals attenuate with increasing depth, leading to a decrease in the signal-to-noise ratio and preventing the accurate acquisition of dynamic signals for the liquid level depth.

The echo signals from the subsurface (including wellhead waves, coupling waves and liquid level waves) contain complex noise signals and exhibit both periodic and random characteristics. The periodic noise in the subsurface is similar to the liquid level wave signals in amplitude and frequency, introducing uncertainty in the identification of the liquid level wave signals [34]. To accurately identify the position of the liquid level wave, it is necessary to effectively process the noise signals, which is mainly divided into two aspects. First, the periodic noise signals from the well must be addressed by identifying the pattern of periodic changes and eliminating it. Then, for the random noise signals, the superposition and processing of the acoustic signals should be carried out.

#### 3.2.1. Periodic Noise Signal Suppression Algorithms

##### Periodic Noise Signal Filtering

To eliminate the interference of high-frequency components in the underground acoustic signals, this algorithm employs a Butterworth bandpass filter to filter the signals. The underground acoustic signals contain not only liquid level waves but also wellhead waves, coupling waves and other components, with the wellhead wave frequency ranging from 40 Hz to 50 Hz, that of coupling waves ranging from 10 Hz to 20 Hz and that of liquid level waves ranging from 0 Hz to 10 Hz, and the frequency of the periodic underground noise also lies within 0 Hz to 10 Hz. Therefore, the chosen bandpass filter has a cutoff frequency of 0 Hz to 10 Hz. The filtered waveform is shown in Figure 8.

##### Identification and Suppression of Periodic Noise Signals

The liquid surface echo signal is within 5000 data points, and the signals within the first 5000 points contain both valid liquid surface echo signals and periodic noise signals. These noise signals bear a strong resemblance to the liquid level echo signals in both frequency and amplitude. To prevent interference from the liquid level echo signal in the identification of the noise signals, the calculation for the recognition of periodic noise begins from the tail end of the collected signal.

Drawing from engineering experience, the number of data points between the peaks of the periodic underground noise signals is between 4000 and 7000. Consequently, starting from the last point of the acoustic signal (the 22,500th point), 7000 points are extracted in the forward direction, and the point of the maximum signal amplitude, referred to as Max1, is identified. Starting from the point of the maximum value, Max1, another 7000 points are extracted in the forward direction to identify the point with the maximum amplitude, known as Max2. A threshold α is then set, where α=min(Max1,Max2)/max(Max2,Max1). This threshold can be adjusted based on the actual conditions of the well. Additionally, considering the potential variations in the amplitude values of the noise signal peaks, if the threshold is greater than or equal to 0.85, it is considered that Max1 and Max2 represent the peak values of the noise signal. The amplitude value of the noise can be set to the average of Max1 and Max2, as illustrated in Figure 9.
(3)Max=(Max1+Max2)/2

From the beginning of the collected signal time series at point 0 to the end at point 22,500, the amplitude values of the signal are compared with the Max value. To accurately and effectively identify the amplitude of the peaks, an amplitude deviation value of 150 is set. Amplitude values that are less than Max − 150 or greater than Max + 150 are set to 0. Additionally, since the signal is sampled at every 4 milliseconds for one point and the period of the noise is generally over 4000 points, an extreme value is taken every 400 points, and so on. All of the extreme values are placed into the sequence MaxList, and the sequence indices corresponding to these extreme values are stored in the sequence MaxListIndex. The index values and amplitude values of each extreme point are recorded, as shown in Figure 10.

We traverse the MaxListIndex = [a, b, c, d, e, f, g, h, i, j, k, l] and calculate the differences in indices between any two points in the MaxListIndex sequence. We record the points with differences within the range of 4000 to 7000 into the MaxMapping and determine whether the differences in the MaxMapping are within an error range of 200. We retain the indices corresponding to differences of less than 200, resulting in the following sequence:

A = [187,6871, 13,379, 20,191];

B = [187, 6651, 13,379, 20,191];

C = [6345, 9848, 13,379, 16,906, 20,432].

Sequences A, B and C are shown in Figure 11, Figure 12 and Figure 13, respectively. By summing the indices of the sequences, the sequence with the largest sum is determined to be sequence C [a, g, h, j, l]. Here, point a represents a noise cycle point before the device is excited and does not participate in the calculation of the liquid level wave depth. Therefore, only points g, h, j and l are marked in the original signal (as shown in Figure 14), and the amplitude value corresponding to point g within the range of 0 to 5000 points is set to 0. This effectively removes the extreme value of the periodic noise, preventing the noise signal point from being mistakenly identified as a liquid level wave signal point.

#### 3.2.2. Random Noise Signal Suppression Algorithms

##### Random Noise Signal Filtering

In addition to periodic noise, the collected underground acoustic signals also contain a significant amount of random noise, such as knocking from the wellhead Christmas tree, ground construction, impacts from the nodding donkey (pole pumping unit) and noise from passing vehicles. Based on the practical engineering experience of the oil extraction plant, the frequencies of this random noise mostly lie within the range of 0 Hz to 5 Hz. To effectively extract the random noise, a Butterworth bandpass filter is applied to the collected acoustic signals. The chosen cutoff frequencies are set from 0 Hz to 5 Hz, and the filtering results are shown in Figure 15.

##### Identification and Suppression of Random Noise Signals

(1)Upon the analysis of the collected acoustic signals, it was observed that random noise signals exhibit multiple extreme points within a half period. This characteristic does not align with the features of liquid level echo signals. Therefore, a derivative operation is performed on the time series to identify these multiple extreme points within the half period of the collected signal. The amplitude values of the signal at these points are then set to 0 to eliminate the outliers, as shown in Figure 16.(2)After the processing in step (1), there will be waveforms with amplitudes of 0 within a period, which do not conform to the characteristics of liquid level echo signals. Therefore, the amplitude values of abnormal waveforms that are not sine waves are all set to 0, as shown in Figure 17.(3)For excitation signals and liquid level echo signals, the wavelength of the liquid level wave in the latter half of the period should be longer than that in the first half. Consequently, the amplitude values of the sine wave where the wavelength in the latter half of the period is shorter than that in the first half are set to 0, as illustrated in Figure 18.(4)We calculate the sine wave with the greatest peak-to-trough difference within the signal, which is identified as the liquid level echo signal, as shown in Figure 19.(5)As shown in Figure 20, the collected signals are first processed to suppress periodic and random noise and then superimposed three times, resulting in a significant enhancement in the amplitude of the liquid level echo signal, while also effectively suppressing the noise signals. However, as shown in Figure 21, the original signals without noise suppression are superimposed three times, and the difference in amplitude between the liquid level echo and the noise is not distinct, making it difficult to differentiate between the liquid level echo and the noise signals. This demonstrates that the noise reduction method effectively suppresses random and periodic noise signals, thereby effectively improving the recognition effect of the liquid level echo signal.

### 3.3. Visualization Software

In the field, for the real-time monitoring of oil well dynamic liquid levels, first, it is necessary to ensure that there is no issue with the real-time communication between the wellhead detection device and the intelligent terminal. Then, by pressing the “Start Detection” button on the software interface, the measurement of the liquid level depth commences. The results of the operation are depicted in Figure 22.

Then, as shown in Figure 23, users can utilize the historical data preservation and viewing functions to query and analyze the historical data of the dynamic liquid level depth. Additionally, they can also observe the trends of changes in the liquid level depth.

## 4. Experimentation and Evaluation


*Installation and Operation*


The field experiments are conducted at the Jianghan Oilfield. Firstly, the wellhead detection device is connected to the oil extraction casing using the reserved threaded interface so as to ensure the connection’s security. The wellhead detection device and the on-site installation are shown in Figure 24. The data are collected by connecting the detection device to the intelligent terminal equipment via a serial cable. Then, after confirming that there are no issues with the electrical connections, the dynamic liquid level detection software is launched. Some commands are sent through the intelligent terminal device to check the data upload status. The real-time data transmission interface is depicted in Figure 25.

From March to October 2023, measurements were conducted on the oil wells numbered 76-2, 76-7-2, 76-5-8, 76-6-5, 76-4-1, SK8-17, SK8-11 and SH6-P20 at the Jianghan Oilfield. The measurement results were then compared with the actual liquid level depth values. Table 1 presents the results of the liquid level echo before noise reduction processing, and Table 2 presents the results after noise reduction processing. The test results indicate that the error of the measured values compared to the actual values is less than 1.2%.

Multiple on-site tests have demonstrated that the oil well dynamic liquid level monitoring system operates stably, with a wide range of liquid level depths and high precision. The wellhead detection device is designed with a low cost and features an innovative infrasound emission method, ensuring a high safety factor. The dynamic liquid level detection software is comprehensive, integrating algorithms for the noise reduction of the acoustic signals, the automatic calculation of the sound speed and liquid level depth and other signal processing functions. It offers features such as automatic data storage, historical record retrieval and the analysis of liquid level depth trend changes. The oil well dynamic liquid level monitoring system is composed of a wellhead detection device and intelligent detection equipment. The system possesses a complete set of capabilities, including automatic infrasound excitation, automatic data acquisition and upload, the high-precision autonomous analysis of acoustic signals and data visualization for dynamic liquid level monitoring.

## 5. Conclusions

This paper presents the design of an IoT-based underground dynamic liquid level monitoring system that leverages a spring-driven excitation and data acquisition device for the efficient collection of acoustic wave data. The system employs advanced filtering techniques to distinguish between wellbore and wellsite noise, effectively isolating the coupling and liquid level waves. By calculating the sound velocity from the coupling wave frequency and integrating a denoising algorithm to pinpoint the liquid level echo, the system achieves precise liquid level depth measurement. Our contributions are highlighted in four key are as.

(1)Innovation in Real-Time Monitoring Algorithms: We have developed a novel noise reduction algorithm tailored to periodic and random noise in oil well dynamic liquid level monitoring. This algorithm enhances the liquid surface echo signal amplitude while reducing background noise, significantly improving the echo recognition accuracy. Compared to traditional methods, our algorithm demonstrates superior performance in practical settings, offering a cutting-edge solution for oil well monitoring advancements.(2)Validation through Comparative Analysis: Our comparative analysis of actual monitoring data indicates that the noise reduction algorithm maintains the monitoring errors within a strict 2% margin. This not only validates the algorithm’s effectiveness but also shows its significant advantage over non-noise-reduction monitoring techniques, providing a robust technical approach to the development of dynamic liquid level monitoring systems.(3)Addressing Algorithm Limitations: While the noise reduction algorithm performs exceptionally in most scenarios, we acknowledge its limitations in high-pressure gas wells and complex underground environments. We are committed to ongoing research to enhance the algorithm’s generalization capabilities, aiming to more effectively differentiate between liquid surface waves and noise, thereby meeting the monitoring demands of challenging environments.(4)Integrating Data-Driven Methods: To tackle the uncertainties in dynamic liquid level depth monitoring, we propose the incorporation of deep learning technology to bolster the algorithm’s generalization and robustness. Deep learning’s capabilities for feature extraction from large datasets offer powerful data processing for monitoring algorithms. Additionally, we will explore transfer learning strategies to address the small sample size issues, expanding the underground dynamic liquid level data sample library and enhancing the system’s adaptability and reliability in diverse environments.

Through in-depth research and technological innovation in these areas, we anticipate significant advancements in the real-time monitoring of oil well dynamic liquid levels, thereby providing robust technical support for the stable and sustainable development of the oil and gas industry.

## Figures and Tables

**Figure 1 sensors-24-03607-f001:**
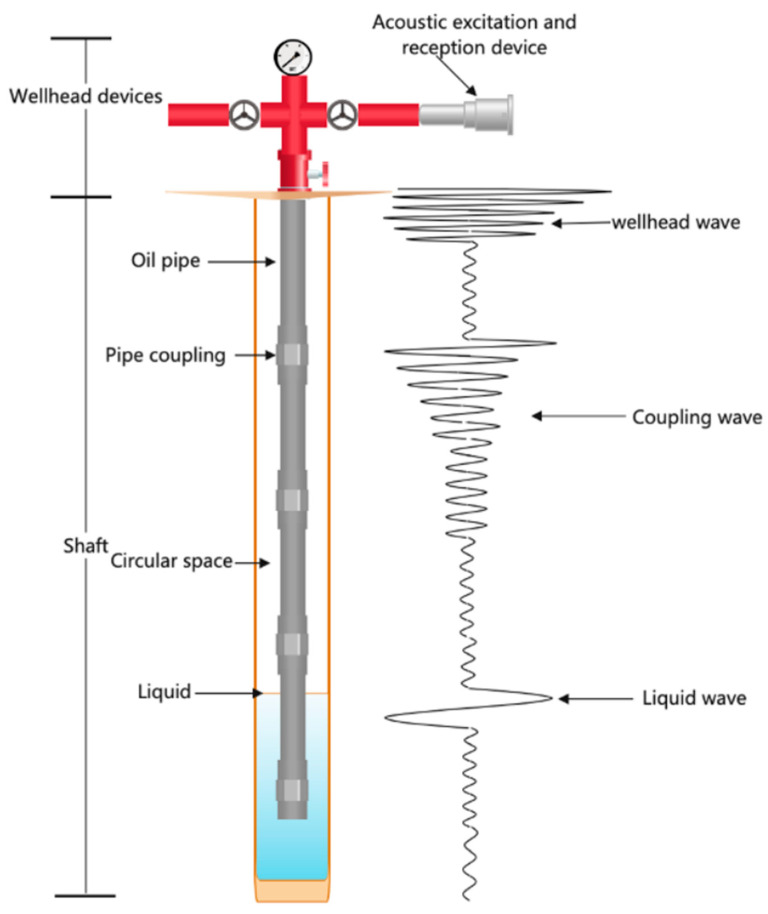
Propagation model of sound waves in the oil tube.

**Figure 2 sensors-24-03607-f002:**
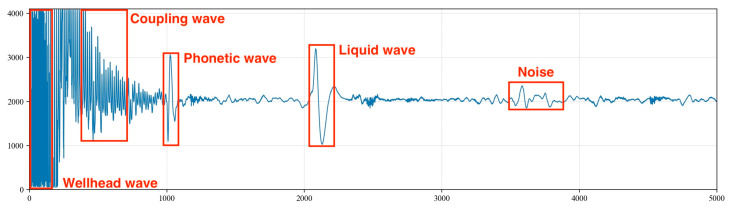
Acoustic waveform and positions of various echoes.

**Figure 3 sensors-24-03607-f003:**
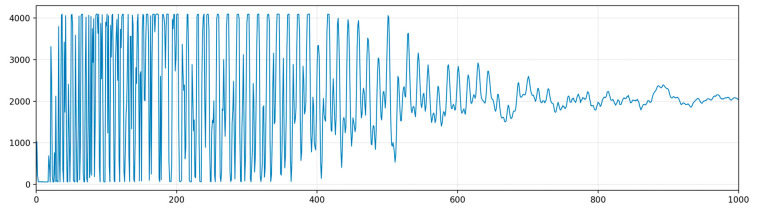
Time domain waveform of the coupling wave.

**Figure 4 sensors-24-03607-f004:**
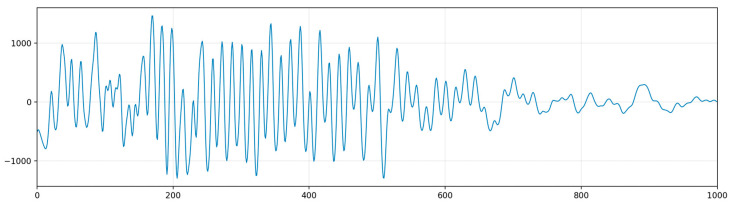
Filtered coupling wave.

**Figure 5 sensors-24-03607-f005:**
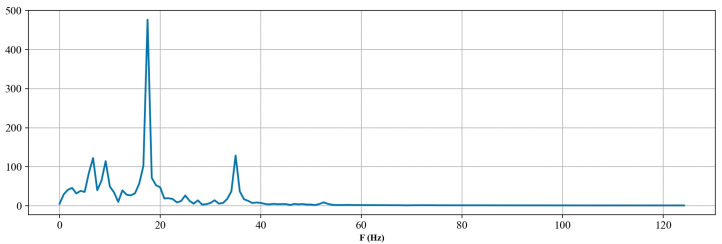
Spectrum of the coupling wave reflection signal.

**Figure 6 sensors-24-03607-f006:**
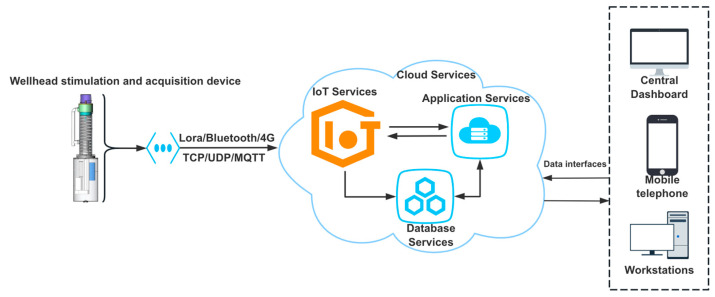
Intelligent dynamic liquid level monitoring and measurement system.

**Figure 7 sensors-24-03607-f007:**
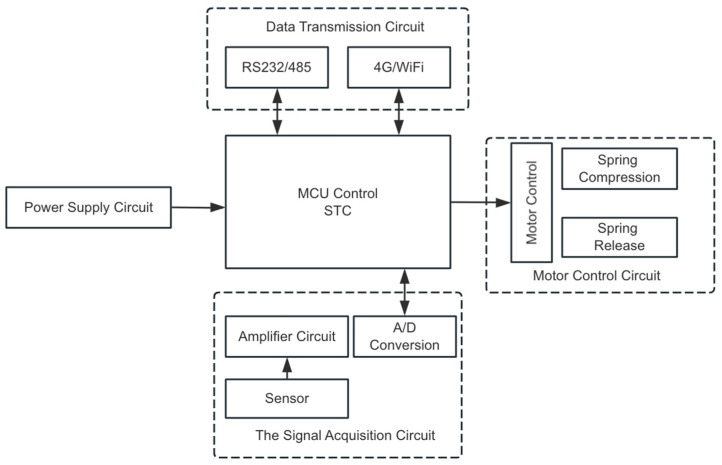
The overall architecture of the wellhead excitation and data acquisition device.

**Figure 8 sensors-24-03607-f008:**
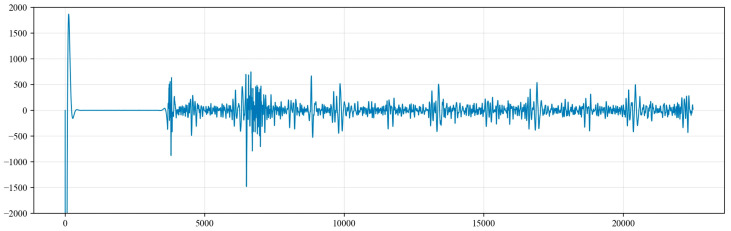
The filtering results after 0 Hz to 10 Hz bandpass filtering.

**Figure 9 sensors-24-03607-f009:**
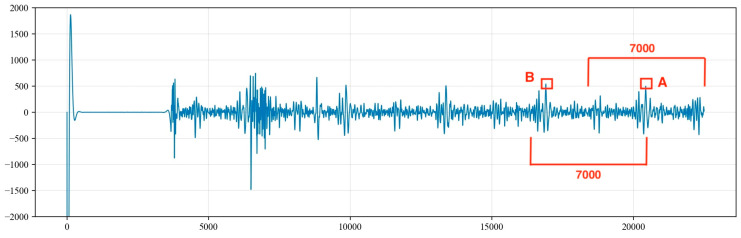
Determination of maximum values for periodic noise signals.

**Figure 10 sensors-24-03607-f010:**
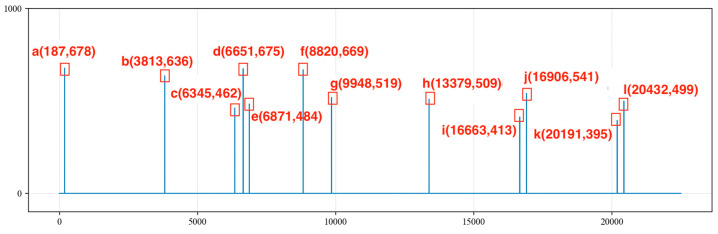
Storing the extreme value points of periodic noise signals.

**Figure 11 sensors-24-03607-f011:**
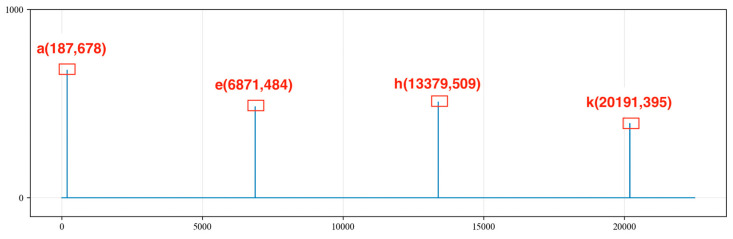
Sequence A of extreme values.

**Figure 12 sensors-24-03607-f012:**
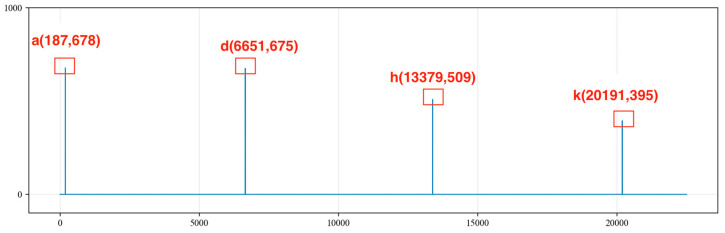
Sequence B of extreme values.

**Figure 13 sensors-24-03607-f013:**
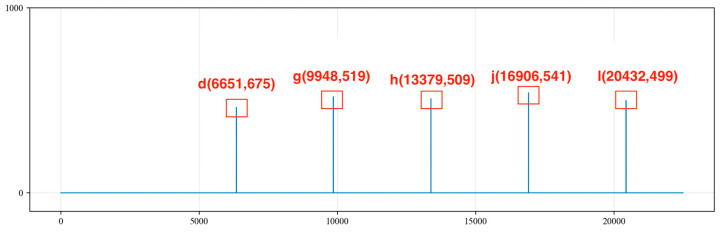
Sequence C of extreme values.

**Figure 14 sensors-24-03607-f014:**
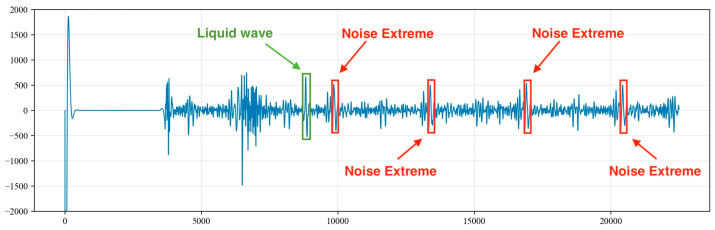
Index positions of extreme noise values in the corresponding original signal.

**Figure 15 sensors-24-03607-f015:**
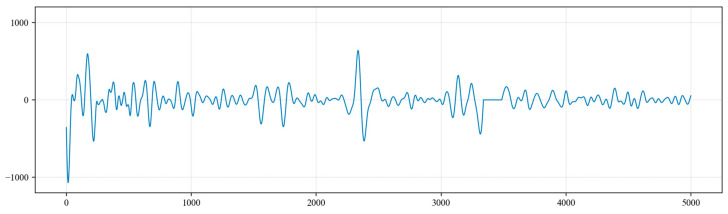
The filtering results after 0 Hz to 5 Hz bandpass filtering.

**Figure 16 sensors-24-03607-f016:**
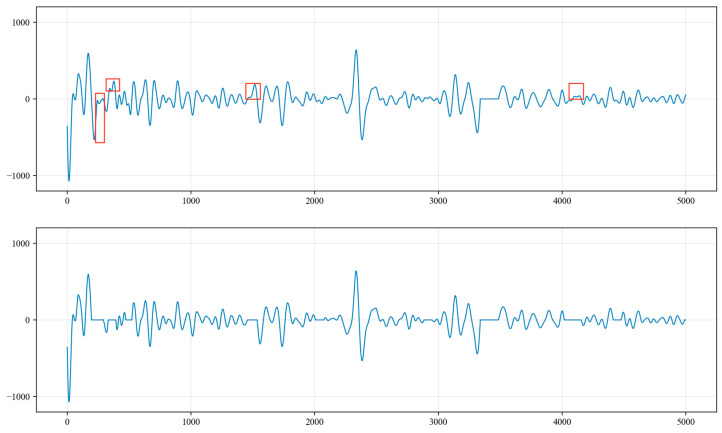
Waveform analysis after removing multiple extreme points in a half period.

**Figure 17 sensors-24-03607-f017:**
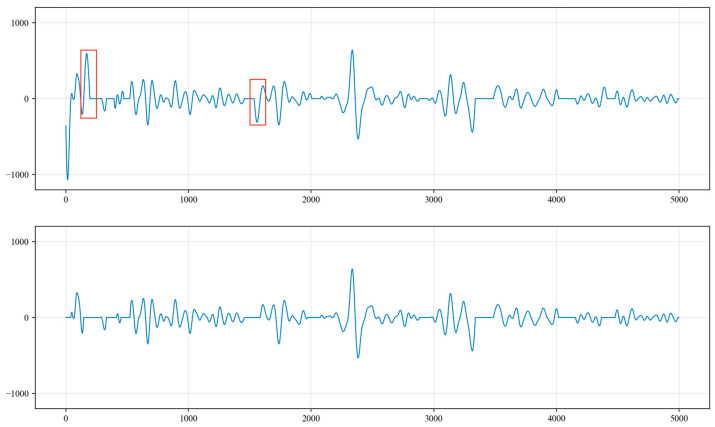
Analysis of abnormal waveforms with non-sinusoidal characteristics.

**Figure 18 sensors-24-03607-f018:**
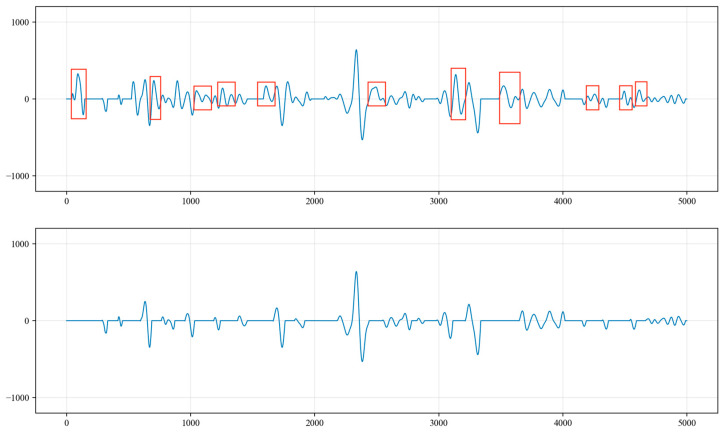
Analysis of cases where the wavelength in the latter half of the period is shorter than that in the first half.

**Figure 19 sensors-24-03607-f019:**
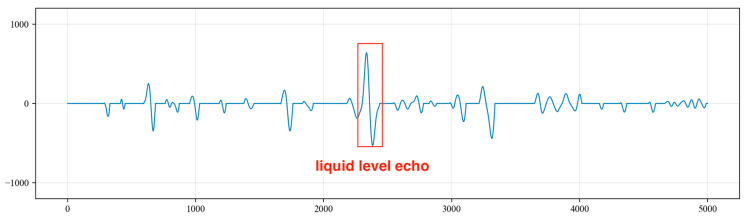
Identification of liquid level echo signal.

**Figure 20 sensors-24-03607-f020:**
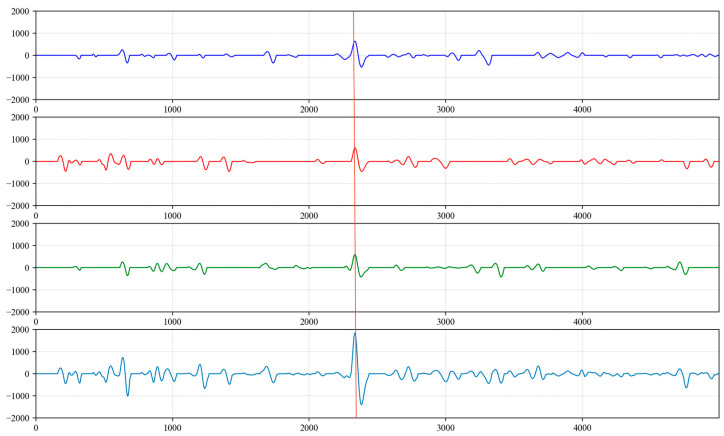
The result of tripling the superimposed signals after noise reduction treatment.

**Figure 21 sensors-24-03607-f021:**
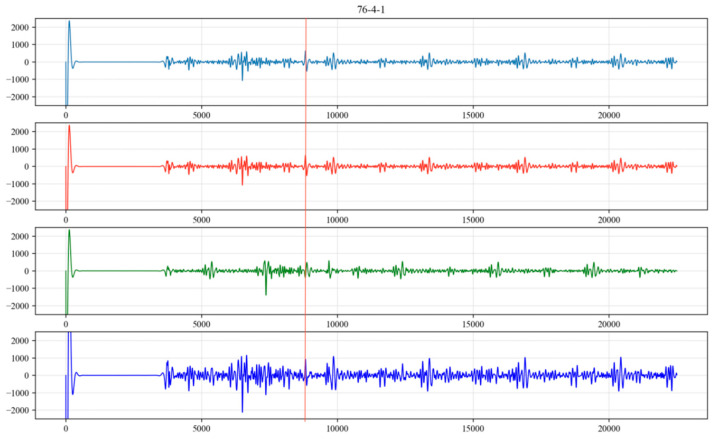
The result of tripling the superimposed original signals without noise reduction.

**Figure 22 sensors-24-03607-f022:**
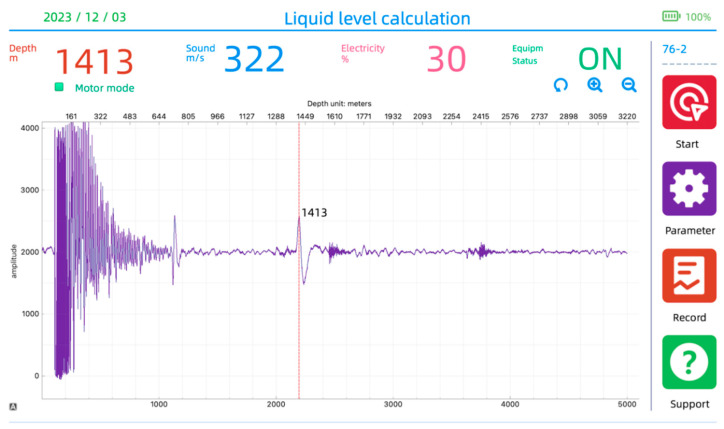
Liquid level depth measurement interface.

**Figure 23 sensors-24-03607-f023:**
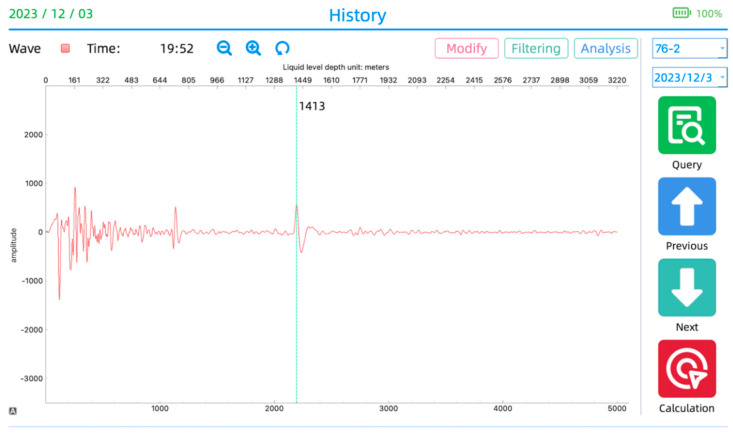
Historical data viewing interface.

**Figure 24 sensors-24-03607-f024:**
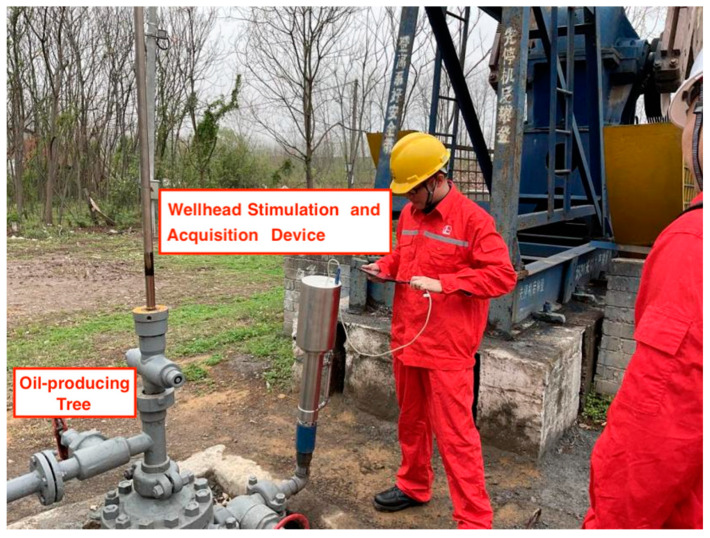
Wellhead detection device and on-site installation at the oilfield.

**Figure 25 sensors-24-03607-f025:**
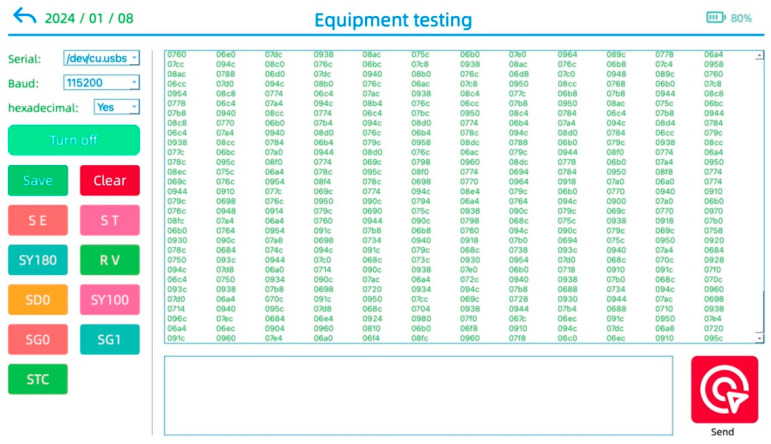
Real-time data communication interface.

**Table 1 sensors-24-03607-t001:** Calculation results of liquid level echo without noise reduction processing.

Well Number	Measured Depth	Actual Depth	Absolute Error	Relative Error
76-2	1240	1340	100	7.46%
76-7-2	676	750	74	9.87%
76-5-8	1233	1545	312	20.19%
76-6-5	1436	1620	184	11.36%
76-4-1	1236	1550	314	20.26%
SK8-17	2075	2240	165	7.34%
SK8-11	2461	2140	321	15.00%
SH6-P20	1922	2430	508	20.91%

**Table 2 sensors-24-03607-t002:** Calculation results of liquid level echo after noise reduction processing.

Well Number	Measured Depth	Actual Depth	Absolute Error	Relative Error
76-2	1343	1340	3	0.22%
76-7-2	754	750	4	0.53%
76-5-8	1537	1545	8	0.51%
76-6-5	1625	1620	5	0.31%
76-4-1	1568	1550	18	1.16%
SK8-17	2246	2240	6	0.27%
SK8-11	2147	2140	7	0.33%
SH6-P20	2438	2430	8	0.33%

## Data Availability

The original contributions presented in the study are included in the article, further inquiries can be directed to the corresponding author.

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
