# Peer review of "Robust Intelligent Monitoring and Measurement System toward Downhole Dynamic Liquid Level"

_sensors, 2024, doi:10.3390/s24113607_

Round 1
Reviewer 1 Report
Comments and Suggestions for Authors
This paper designed a set of underground dynamic liquid level depth monitoring system based on Internet of Things (IoT). The topic is interesting but need major revisions as follows,
1. In the introduction, the author's name should be given by their surname, not their full name.
2. At the end of the introduction, the article's framework structure needs to be explained.
3. All tables should be in a three-line format.
4. Most importantly, the newly designed monitoring system lacks uncertainty analysis. In physical experiments, uncertainty analysis, or experimental uncertainty assessment, deals with assessing the uncertainty in a measurement. An experiment designed to determine an effect, demonstrate a law, or estimate the numerical value of a physical variable will be affected by errors due to instrumentation, methodology, presence of confounding effects, and so on. Experimental uncertainty estimates are needed to assess confidence in the results.
Moderate editing of English language required
Reviewer 2 Report
Comments and Suggestions for Authors
This research introduces an intelligent monitoring and measurement system for oil well dynamic liquid levels and validates its accuracy and reliability using actual oil field test data. The overall structure of the paper is clear, and the expression is fluent, making it easy for readers to follow. By addressing practical oil well production scenarios, it effectively tackles practical issues. In summary, it is a study with practical application value. I recommend minor modifications for publication.
1. In the introduction section, it is recommended to provide a comparative analysis between the system proposed in this study and other dynamic liquid level monitoring and measurement technologies discussed in the second paragraph. This comparison should highlight the strengths and advantages of the research system, thereby providing clarity to the reader regarding its innovative contributions to the field.
2. In the introduction section, the main contributions of the papers can be directly expressed in paragraph form, eliminating the need for additional listing formats as shown from Line 82 to Line 92.
3. In the introduction section, it would be beneficial to include additional information stating that the system under study has been successfully implemented in the Jianghan Oilfield, yielding desirable results. This further underscores the practical application value of the research.
Reviewer 3 Report
Comments and Suggestions for Authors
The article titled "Robust Intelligent Monitoring and Measurement System toward Downhole Dynamic Liquid Level" stands out for its timely insights and original findings. Although the article may present a relevant contribution to the scientific community, it is still not entirely clear from the reader's perspective its core contributions.
The positive issues of the article, such as the authors' use of illustrative figures are quite commendable. However, readers may find the lack of detailed explanation and description somewhat limiting in fully grasping the content. To address this issue, I propose restructuring the conclusions section into distinct subsections:
1. Contributions to Theory: This section should highlight the novel insights and advancements introduced by the article, emphasizing how it extends the existing theoretical framework.
2. Practical Implications: The authors can show the practical relevance of their findings, outlining their implications for industry practitioners and stakeholders.
3. Methodological Limitations: Here the authors should acknowledge the constraints and limitations of the research methodology.
4. Future Directions: The authors should offer suggestions for future research directions to foster scholarly dialogue.
While some of these aspects may already be addressed to some extent in the current conclusions section (section 5), expanding the contributions in a structured manner will provide readers with a more comprehensive understanding of the article's significance and implications.
Good luck with your revision and I hope that my comments help to build a more solid and in-depth article.
Comments on the Quality of English LanguageMinor editing of English language required.
Round 2
Reviewer 1 Report
Comments and Suggestions for Authors
The authors have modified the manuscript according to comments. It is recommended to be published in the current form.
Comments on the Quality of English LanguageMinor editing of English language required
Reviewer 3 Report
Comments and Suggestions for Authors
The authors replied to my comments.
Comments on the Quality of English LanguageMinor editing of the English language is required.